# Adverse Childhood Experiences Predict the Phenome of Affective Disorders and These Effects Are Mediated by Staging, Neuroimmunotoxic and Growth Factor Profiles

**DOI:** 10.3390/cells11091564

**Published:** 2022-05-07

**Authors:** Michael Maes, Muanpetch Rachayon, Ketsupar Jirakran, Pimpayao Sodsai, Siriwan Klinchanhom, Monojit Debnath, Agnieska Basta-Kaim, Marta Kubera, Abbas F. Almulla, Atapol Sughondhabirom

**Affiliations:** 1Department of Psychiatry, Faculty of Medicine, King Chulalongkorn Memorial Hospital, Chulalongkorn University and The Thai Red Cross Society, Bangkok 10330, Thailand; muanpetch.mp@gmail.com (M.R.); ket.kett@hotmail.com (K.J.); abbass.chem.almulla1991@gmail.com (A.F.A.); atapol.s@gmail.com (A.S.); 2IMPACT Strategic Research Center, Barwon Health, Geelong 3220, Australia; 3Department of Psychiatry, Medical University of Plovdiv, 4000 Plovdiv, Bulgaria; 4Maximizing Thai Children’s Developmental Potential Research Unit, Department of Pediatrics, Faculty of Medicine, Chulalongkorn University, Bangkok 10330, Thailand; 5Center of Excellence in Immunology and Immune-Mediated Diseases, Department of Microbiology, Faculty of Medicine, Chulalongkorn University, Bangkok 10330, Thailand; yokpim@gmail.com (P.S.); siriwanklinchanhom@gmail.com (S.K.); 6Division of Immunology, Department of Microbiology, Faculty of Medicine, Chulalongkorn University, Bangkok 10330, Thailand; 7Department of Human Genetics, National Institute of Mental Health and Neurosciences, Bangalore 560 029, India; monozeet@gmail.com; 8Department of Experimental Neuroendocrinology, Maj Institute of Pharmacology, Polish Academy of Sciences, 31-343 Kraków, Poland; basta@if-pan.krakow.pl (A.B.-K.); kubera@if-pan.krakow.pl (M.K.); 9Medical Laboratory Technology Department, College of Medical Technology, The Islamic University, Najaf 54001, Iraq

**Keywords:** early lifetime trauma, depression, mood disorders, inflammation, neuroimmune, cytokines, psychiatry

## Abstract

Adverse childhood experiences (ACEs) enhance pro-inflammatory and pro-oxidant responses. In affective disorders, recent precision nomothetic psychiatry studies disclosed new pathway phenotypes, including an ROI—reoccurrence of illness (ROI)—oxidative stress latent construct. The aim of the present study is to delineate a) whether ACEs sensitize the M1 macrophage, the T helper cells (Th)1, Th2, and Th17, the IRS (immune-inflammatory-responses system), the CIRS (compensatory immunoregulatory system), and the neuroimmunotoxic and growth factor (GF) profiles and whether they are associated with ROI and the phenome of affective disorders and b) the molecular pathways underpinning the effects of the ACEs. We collected supernatants of stimulated (5 μg/mL of PHA and 25 μg/mL of LPS) and unstimulated diluted whole blood in 20 healthy controls and 30 depressed patients and measured a panel of 27 cytokines/GF using a Luminex method. ACEs (comprising mental and physical trauma, mental neglect, domestic violence, family history of mental disease, and parent loss) are accompanied by the increased stimulated, but not unstimulated, production of M1, Th1, Th2, Th17, IRS, neuroimmunotoxic, and GF profiles and are strongly correlated with ROI and the phenome. A latent vector extracted from the ROI features (recurrent episodes and suicidal behaviors) and the IRS/neuroimmunotoxic/GF profiles explains 66.8% of the variance in the phenome and completely mediates the effects of ACEs on the phenome. Enrichment analysis showed that the ACE-associated sensitization of immune/GF profiles involves JAK-STAT, nuclear factor-κB, tumor necrosis factor-α, G-protein coupled receptor, PI3K/Akt/RAS/MAPK, and hypoxia signaling. In summary, the ACE-induced sensitization of immune pathways and secondary immune hits predicts the phenome of affective disorders.

## 1. Introduction

Affective disorders progress through distinct lifetime epochs, which include adverse childhood experiences (ACEs), recurrent depressive episodes with or without (hypo)manic episodes, and recurrent suicidal behaviors that alternate with euthymic phases and a residual stage marked by functional impairments and neurocognitive deficits [1,2,3]. The accumulation of many traumatic experiences throughout childhood, such as physical and emotional neglect or abuse, sexual abuse, family strife, and bullying, is connected with the later development of depression and bipolar disorder (BD), disease intensity, increased suicidal behaviors, co-occurring anxiety disorders, and impairments in verbal fluency and executive functions [1,4,5,6,7,8,9,10,11,12,13,14,15,16,17,18,19,20]. Additionally, during episodes of unipolar or bipolar major depressive disorder, named major depressive episodes (MDEs), the cumulative effects of ACEs predict the type of treatment received, such as, specifically, the use of mood stabilizers, lithium, and antipsychotics, and they predict socioeconomic status, including income, increased disabilities, and a decreased health-related quality of life (HR-QoL) [1,21,22,23,24,25]. It is important to note that women report more ACEs than men [1,26,27], suggesting that girls are more likely to develop affective disorders and more severe psychopathology than boys when subjected to ACEs [25]. As such, the cumulative effects of ACEs impact the phenome of affective disorders, which is composed of symptomatome (aggregate of symptoms and co-morbidities) and phenomenome (aggregate of the self-experience of the illness) features [1,2,3,28,29].

ACEs have an effect on the course and progression of illness and may result in more recurrent episodes [1,3,4,5] of major depression and (hypo)mania, whilst the number of episodes and suicide attempts is associated with increased symptomatic severity and increased risk of future episodes and functional decline and an increased risk of neurocognitive impairments [1,2,3]. Precision medicine techniques showed that the cumulative effects of diverse ACEs are strongly associated with a new recurrence of illness index (ROI), conceptualized as a latent vector extracted from the number of depressed and hypomanic episodes, as well as lifetime suicide attempts and ideation [1,3]. Furthermore, we were able to extract one replicable latent vector from the above ROI indicators and the total ACE score, generating an ACE–ROI index, which strongly predicts the phenome of affective disorders [2,3,25]. As such, the effects of ACEs on the phenome of affective disorders are, at least in part, mediated by the ROI [1,2,25].

Major depressive disorder (MDD) and BD are characterized by the activation of immune-inflammatory, oxidative, and nitrosative stress (IO&NS) pathways [30,31]. Firstly, both MDD and BD are linked with elevated reactive oxygen and nitrogen species (RONS) and nitro-oxidative stress toxicity (OSTOX), as indicated by lipid peroxidation, aldehyde formation, protein oxidation, and reduced antioxidant defenses [30,31]. Secondly, MDD and BD are characterized by activation of the immune-inflammatory response system (IRS), the compensatory immunoregulatory system (CIRS), which prevents hyperinflammation, and growth factors that further boost the immunological response [32,33]. In affective disorders, IRS is defined by the activation of M1 macrophages, including increased interleukin (IL)-1β, IL-6, IL-8 and tumor necrosis factor (TNF)-α production, T helper (Th)1 cells, including elevated IL-2, IL-12, and interferon (IFN)-γ production, and Th17 (IL-17) phenotypes. CIRS is defined by the activation of Th2 (IL-4, IL-5) and T regulatory (IL-10) cells [32]. The important growth factors associated with depression are platelet-derived growth factor (PDGF), vascular endothelial growth factor (VEGF), and fibroblast growth factor (FGF) [33].

OSTOX, depleted antioxidant defenses, the IRS/CIRS production ratio, a neuroimmunotoxic profile consisting of M1, Th1, and Th17 cytokines, and the growth factors all predict a significant portion of the variance in the phenome of affective disorders and are strongly associated with ROI to the extent that they form the ROI-REDOX (OSTOX, paraoxonase 1) and ROI-IMMUNE (neuroimmunotoxic and growth factors) pathway phenotypes (latent constructs) [1,2,3,25,34], indicating that the redox and immune pathways are intimately linked with ROI and together predict the phenome of affective disorders.

Additionally, there is some evidence that ACEs may operate as sensitizers of the IO&NS pathways [1,2,3,25]. Firstly, physical neglect is related with markers of lipid peroxidation, nitro-oxidative stress, and lipid and protein oxidation in mood disorders, whilst sexual abuse is associated with decreased antioxidant levels, including zinc, albumin, and the thiol (-SH) groups [27]. OSTOX seems to be partly driven by ACEs, particularly physical neglect, and both ACEs and RONS/OSTOX predict the phenome of affective disorders, comorbidity between depression and anxiety disorders, and worse HR-QoL [1,2,3,25,27,34]. Additionally, the effects of ACEs on the phenome of affective disorders are mediated in part by ROI and decreased antioxidant defenses [2,25]. Individuals with a history of multiple ACEs also exhibit signs of activated immune-inflammatory pathways, such as elevated blood levels of inflammatory markers, including C-reactive protein (CRP), IL-6, and the soluble urokinase plasminogen activator receptor [12,35,36,37,38,39,40,41,42]. Lo Lacono et al. [43] reported that altered levels of VEGF in depression were independent of the effects of low parental care. Nonetheless, there are no data on whether ACEs are associated with enhanced M1, Th1, Th17, and IRS or with neuroimmunotoxic profiles, increased growth factor production, and diminished CIRS profiles, or whether the effects of ACEs on the phenome of affective disorders are mediated by ROI.

Hence, the current study was conducted to examine whether (a) ACEs are significantly and positively associated with increased M1, Th1, Th17, neuroimmunotoxic, IRS/CIRS, and growth factor profiles; (b) the effects of ACEs on the phenome of affective disorders are mediated by ROI; and (c) a common core underpins ACEs, ROI, immune profiles, and the phenome of affective disorders. Toward this end we employed a precision nomothetic approach [3], including Partial Least Squares analysis to delineate the causal effects of ACEs on ROI and immune activation and the cumulative effects of those predictor variables on the affective phenome. The precision nomothetic approach [3] was also used to delineate new pathway phenotypes [3] based on ACEs, ROI, immune activation, and the affective phenome.

## 2. Methods and Participants

### 2.1. Participants

In this research, we included 20 healthy controls and 30 depressed patients recruited from the outpatient clinic of the Department of Psychiatry at King Chulalongkorn Memorial Hospital in Bangkok, Thailand. The controls were recruited by word of mouth within the same catchment region, namely Bangkok. We recruited participants of both sexes, aged 18 to 65 years. The patients were diagnosed with MDE, according to DSM-5 criteria, and had moderate to severe depression, as determined by the Hamilton Depression Rating Scale (HDRS) [44]. The exclusion criteria for patients were other DSM-5 axis 1 disorders, including obsessive compulsive disorder, psycho-organic disorders, schizophrenia, schizoaffective disorders, substance abuse disorders, and post-traumatic stress disorder. Controls were excluded from the study if they had had any DSM-5 axis 1 diagnosis. Both the patients and the controls were excluded if they (a) had (auto)immune diseases, such as cancer, chronic obstructive pulmonary disease, type 1 diabetes, psoriasis, and inflammatory bowel disease; (b) had neurological disorders, including neurodegenerative and neuroinflammatory disorders (e.g., epilepsy, multiple sclerosis, stroke, or Alzheimer’s or Parkinson’s disease; (c) had previous treatments with immunomodulatory drugs; (d) had allergic or inflammatory reactions three months prior to the study; (e) had treatments with therapeutic doses of antioxidant and omega-3 supplements, or anti-inflammatory medication one month prior to the study; or (f) were pregnant or lactating women. We statistically accounted for the potential effects of the patients’ medication use, including sertraline (*n* = 18), various antidepressants (*n* = 8, including escitalopram, fluoxetine, bupropion, venlafaxine, and mirtazapine), mood stabilizers (*n* = 4), atypical antipsychotics (*n* = 14), and benzodiazepines (*n* = 22).

The study was conducted in accordance with international and Thai ethical standards and privacy laws. The Institutional Review Board of Chulalongkorn University’s Faculty of Medicine in Bangkok, Thailand (#528/63), authorized the research. All the controls and patients provided written informed consent before participation in this research.

### 2.2. Clinical Measurements

Semi-structured interviews were conducted by a research assistant specialized in mood disorders. An experienced psychiatrist administered the HDRS, a 17-item version, to evaluate the severity of the depressive symptoms [44] and the Mini-International Neuropsychiatric Interview to make the axis-1 diagnoses [45]. The ACEs were assessed using the Adverse Childhood Experiences (ACE) Questionnaire [46], which comprises 28 items, scoring 10 domains, namely (1) mental trauma, (2) physical trauma, (3) sexual abuse, (4) mental neglect, (5) physical neglect, (6) witnessing a mother being abused (domestic violence), (7) family member with drug abuse, (8) family member with depression/mental illness, (9) losing a parent to separation, death, divorce, and (10) a family member who is in prison. Anxiety levels were assessed using the Thai state version of the State-Trait Anxiety Assessment (STAI) [47].

To calculate the ROI, we counted the number of depressed and (hypo)manic episodes, as well as recent and lifetime suicidal behaviors (SB), using the Columbia-Suicide Severity Rating Scale (C-SSRS) lifeline version [48]. Recent suicidal behaviors were computed as the first principal component (PC) (labeled “PC recent SB”), extracted from nine C-SSRS items, “namely wish to be dead, non-specific active suicidal thoughts, active suicidal ideation with any methods, active suicidal ideation with some intent to act, active suicidal ideation with specific plan/intent, frequency and duration of suicidal ideation, actual attempts, and total number of actual attempts (all past month)” [34]. This first PC explained 60.54% of the variance, and the nine items were highly loaded on this PC (>0.6) [34]. Lifetime SB was computed as a principal component (PC) (labeled “PC lifetime SB”) “extracted from 11 C-SSRS items, namely lifetime wish to die, non-specific active suicidal thoughts, active suicidal ideation with any methods, active suicidal ideation with some intent to act, active suicidal ideation with specific plan/intent, frequency and duration of ideation, number of actual attempts, preparatory acts or behavior, and total number of preparatory acts (all lifetime)” [34]. This first PC explained 62.21% of the variance and all 11 items showed loadings >0.740. The ROI was conceptualized as the first latent vector (LV) extracted (by mean of factor analysis) from the total number of episodes, the number of depressive episodes, the PC lifetime SB, the lifetime suicidal ideation, and the number of lifetime suicidal attempts; this LV explained 75.6% of the variance, with the loadings being >0.664 and with adequate psychometric properties [34]. The phenome of depression was computed by extracting the first LV from the total HDRS and STAI scores and the PC recent SB; the diagnosis was rated as 0 for controls, 1 for simple MDD/MDE, and 2 for MDD/MDE with psychotic/melancholia features. This LV showed excellent psychometric properties and loadings >0.9 on all indicators [34].

The body mass index (BMI) was computed as body weight (in kg) divided by length squared (in meter). The diagnosis of tobacco use disorder (TUD) was made using the DSM-5 criteria.

### 2.3. Assays

Blood was taken in BD Vacutainer^®^ EDTA (10 mL) tubes at 8:00 a.m., after an overnight fast (at least 10 h) (BD Biosciences, Franklin Lakes, NJ, USA). We quantified the cytokines/chemokines/growth factors in unstimulated and stimulated diluted whole blood culture supernatant [49,50,51]. Whole blood culture supernatants, both stimulated and unstimulated, were used because this method allows the assay of cytokines or growth factors which are otherwise difficult to measure in serum or plasma, including IL-5, IFN-γ, IL-2, and IL-15. Moreover, lipopolysaccharide (LPS) + phytohemagglutinin (PHA)-stimulated cultures were used because these measurements reflect the in vivo cytokine production [49,50,51]. Moreover, the LPS+PHA-stimulated production of cytokines and growth factors reflects the capacity to respond to polyclonal activators, reflecting the responsivity of the immune system to bacterial and viral infections [49,50,51]. We utilized RPMI-1640 medium, supplemented with L-glutamine and phenol red and containing 1% penicillin (Gibco Life Technologies, USA), with or without 5 µg/mL PHA (Merck, Germany) + 25 µg/mL lipopolysaccharide (LPS; Merck, Germany). On 24-well sterile plates, 1.8 mL of each of these two mediums was mixed with 0.2 mL of 1/10 diluted whole blood. The specimens from each individual were divided into unstimulated and stimulated groups and incubated for 72 h at 37 °C, 5% CO_2_ in a humidified atmosphere. After incubation, the plates were centrifuged at 1500 rpm for 8 min. The supernatants were extracted carefully under sterile conditions, divided into Eppendorf tubes, and immediately frozen at −70 °C until thawed for the cytokine/growth factor assays. The cytokines/growth factors were measured using the Bio-Plex Pro human cytokine 27-plex assay kit (BioRad, Carlsbad, California, United States of America). In brief, the supernatants were diluted fourfold with the medium and incubated with linked magnetic beads for 30 min. After 30 min and 10 min, respectively, the fluorescence intensities (FI) of the detecting antibodies and streptavidin-PE were assessed by the LUMINEX 200 equipment (BioRad, Carlsbad, California, United States of America). We opted to conduct statistical analyses on the fluorescence intensity (FI) values (with the blank analyte removed) in the present study as FI values are often a better alternative than absolute concentrations, especially when several plates are used [52]. All the samples of cytokines were measurable, except for IL-7, which had an unusually large number of results below the assay’s sensitivity (60%) and was therefore omitted from the analyses. IL-13 showed that 30% of the assays had values below the detection limit and, hence, was included. For all investigations, the intra-assay coefficient of variation values were less than 11%. Appendix A of the Electronic Supplementary File (ESF) contains the names, acronyms, and official gene symbols for all the cytokines/chemokines/growth factors quantified in this investigation. ESF Appendix A summarizes the different immunological profiles examined in this study.

### 2.4. Statistical Analysis

ANOVA was used to compare scale variables, whereas the chi-square or Fisher’s Exact Probability Test was employed to compare nominal variables across the categories. We performed exploratory factor analysis (unweighted least squares) on the 10 ACE items to delineate possible subdomains. Factorability was checked using the Kaiser–Meyer–Olkin test for sample adequacy (which should be greater than 0.6) and Bartlett’s sphericity test. We used varimax rotation to interpret the factors, considering items with loadings >0.4 to have relevance for the constructs. The correlations between two sets of scale variables were computed using Pearson’s product moment or Spearman’s rank order coefficients, while the associations between the scale and binary variables were examined using point-biserial correlation coefficients. The associations between the ACEs and the immunological profiles and cytokines/growth factors were investigated using generalized estimating equations (GEE) methodology. The pre-specified GEE analysis, which employed repeated measures, included fixed categorical effects of time (unstimulated versus stimulated), groups (high ACE versus low ACE patient groups and controls), and time x group interactions, with sex, smoking, age, and BMI as covariates. The immunological profiles were the key outcome variables in the GEE studies, and if these indicated significant outcomes, we looked at the specific cytokines/growth factors. Using the false discovery rate (FDR) *p*-value, the multiple effects of time or group on immune profiles were adjusted [53]. Additionally, we included the patients’ pharmacological status as a predictor in the GEE analysis to exclude the effect of these possible confounding variables on the immune profiles. None of the demographic, clinical, or cytokine/growth factor data evaluated in this study had missing values. We derived marginal means for the groups and time x group interactions and examined differences using (protected) pairwise contrasts (least significant difference at *p* = 0.05). Multiple regression analysis was used to discover the associations between the ACE scores and the phenome, the ROI, or the key immune profiles, while allowing for the effects of other explanatory variables. To this end, we utilized an automated approach with a *p*-to-entry of 0.05 and a *p*-to-remove of 0.06 when assessing the change in R^2^. Multicollinearity was determined by a tolerance and variance inflation factor, multivariate normality by Cook’s distance and leverage, and homoscedasticity by the White and modified Breusch–Pagan tests. The regression analyses’ results were always bootstrapped using 5.000 bootstrap samples, and the latter were reported if the findings were not concordant. All statistical analyses were conducted using IBM SPSS version 28 for Windows. We used two-tailed tests with an alpha of 0.05 threshold (two-tailed). Using a two-tailed test with a significance threshold of 0.05 and assuming an effect size of 0.23 and a power of 0.80 for two groups with about 0.4 intercorrelations, the estimated sample size for a repeated measurement design ANOVA is approximately 30. Using a significance threshold of 0.05 and assuming an effect size of 0.3 and a power of 0.80 for 4 input variables, the estimated sample size for a multiple regression or pathway analysis is approximately 45.

Partial Least Squares path analysis (SmartPLS) [54] was used to determine the causal relationship between the ACEs, ROI, the immune profiles (all input variables), and the phenome of depression (output variable). All variables were entered either as LVs derived from their manifestations or as single indicators. When the inner and outer models met predefined quality criteria, such as (a) the model fit was greater than 0.08 in terms of standardized root mean squared residual (SRMR); (b) the LVs had a high composite reliability (>0.7), Cronbach’s alpha (0.7), and rho A (>0.8) values, with an average variance extracted >0.5; and (c) all LV loadings were greater than 0.6 at *p* < 0.001, a complete PLS analysis was performed on the significant paths. We also ran a Confirmatory Tetrad analysis to make sure the LVs were not misclassified as reflective models. Using the PLS predict and a tenfold cross-validation technique, the model’s prediction performance was tested.

We constructed seed-gene protein-protein interaction (PPI) networks using the differentially expressed proteins (DEPs) that were increased in subjects with ACEs. We created the networks using STRING 11.0 (https://string-db.org, accessed on 28 March 2022) and IntAct (https://www.ebi.ac.uk/intact/, accessed on 28 March 2022). We built zero-order PPIs (comprised solely of seed proteins), a first-order PPI network (using STRING), and enlarged networks, e.g., using OmicsNet (IntAct, accessed on 28 March 2022). STRING was used to visualize the PP interactions; MetaScape (Metascape, accessed on 28 March 2022) to display the enriched ontology clusters colored by cluster IDs; the REACTOME (European Bioinformatics Institute Pathway Database; https://reactome.org, accessed on 28 March 2022) to map the top Reactome biological pathways; and GoNet (dice-database.org) to create graphs including GO keywords and genes. To identify DEP clusters, a Markov Clustering (MCL) analysis was conducted using STRING. STRING and the Network Analyzer plugin for Cytoscape (https://cytoscape.org, accessed on 28 March 2022) were used to examine the topology of the networks. The Network Analyzer was used to define the backbone of the networks as a collection of top hubs (nodes with the largest degree) and non-hub bottlenecks (nodes with the highest betweenness centrality). The following tools were used to examine the PPI networks for enrichment scores and annotated terms: (a) inBio Discover (login/inBio Discover (inbio-discover.com), accessed on 28 March 2022) to establish the disease annotations associated with the enlarged network; (b) OmicsNet (using InAct) to establish GO and PANTHER (www.pantherdb.org/, accessed on 28 March 2022) biological processes; (c) STRING to establish Kegg pathways (https://genome.jp/kegg/, accessed on 28 March 2022) and GO biological processes; (d) Enrichr (Enrichr (maayanlab.cloud)) to delineate the top 10 Elsevier, Kegg, and Wiki (WikiPathways-WikiPathways) pathways, which were visualized using bar graphs made using Appyter (Appyter (maayanlab.cloud, accessed on 28 March 2022); and (e) Metascape to construct molecular complex detection (MCODE) components using the GO, Wiki, and Kegg pathways.

## 3. Results

### 3.1. Sociodemographic Data of Patients Divided According to ACE Scores and Controls

Table 1 shows that there were no significant differences in age, sex, education, and TUD between the controls and the patients. Depressed patients had a somewhat increased BMI and highly elevated HDRS and STAI scores. The ROI and PC lifetime and current SBs, as well as the LV phenome, were significantly higher in the patients than in the controls. The depressed patients showed a higher prevalence of mental trauma, physical trauma, mental neglect, domestic violence, and loss of a parent. Moreover, sexual abuse was significantly higher in the patients than in the controls, while the other three items did not differ between both study samples.

### 3.2. Factor Structure of the 10 ACE Items

Factor analysis showed that the first three factors explained 50.96% of the variance (KMO = 0.639, Bartlett’s χ2 = 150.427, df = 45, *p* < 0.001). The first varimax-rotated factor loaded highly on mental trauma (0.793), physical trauma (0.523), mental neglect (0.698), domestic violence (0.721), family history of mental disease (0.502), and loss of a parent (0.475). The second factor loaded highly on physical neglect (0.585), divorce (0.666), and criminal behaviors (0.771), while only one item scored highly on factor 3, namely sexual abuse (0.903). A second factor analysis conducted on the six items of factor 1 showed a KMO = 0.723, AVE = 48.40%, and loadings > 0.590 on all items and a Cronbach alpha = 0.781. The items belonging to factor 2 (and the single-item factor 3) were not factorizable. Because all items of ACE subdomain 1 were associated with depression, we labeled this ACE subdomain as the “ACE-depression” or ACE-DEP subdomain. Consequently, we computed the sum of these six first factor items to reflect “ACE-DEP” and divided the patient group using a visual binning method into two groups, namely those with lower ACE-DEP (sum subdomain 1 < 3) versus those with scores ≥3. Accordingly, in the statistical analysis we entered the ACE-DEP score and the sexual trauma score, whereas the other items showed a low prevalence and were not useful in the analyses.

### 3.3. Differences in Immune Profiles between Patients with Low/High ACE-DEP Scores and Controls

Table 2 displays the results of the (un)stimulated immune profiles in the patients divided into those with lower versus higher ACE-DEP scores and the healthy controls. The stimulated production was always significantly higher (*p* < 0.001) than the unstimulated production. All group X time interactions for all immune profiles, except the CIRS profile, were significant and remained significant at *p* < 0.044 after *p*-correction for FDR. We could not find any impact of sex, age, TUD, and BMI. We also examined the possible effects of the drug state of the patients on the results shown in Table 2 but could not find any effects, even without FDR *p*-correction.

The GEE analyses showed significant group X time interactions for 16 cytokines/growth factors (see Table 3). The stimulated production of sIL-1RA, IL-5, CXCL8, IL-9, IL-12, IL-15, IL-17, FGF, G-CSF, CXCL10, PDGF, CCL5, TNF-α, and VEGF was significantly greater in patients with higher ACE-DEP as compared with the controls (either in the group or group x time analysis). There was a greater production of IL-2 in subjects with ACE ≥ 3 as compared with controls (*p* = 0.060) and those with ACE < 3 (*p* = 0.018). The production of FGF was significantly higher in participants with ACE-DEP ≥ 3 as compared with patients with ACE-DEP < 3. In any case, the production of these 15 cytokines/chemokines was always significantly increased in participants with ACE-DEP ≥ 3 as compared with all other subjects.

### 3.4. Associations between ACEs and ROI, SBs, and the Phenome

Table 4 displays the correlations between the 10 ACEs and the ACE-DEP score and the features of depression. Thus, there were significant correlations between ROI and mental and physical trauma, mental neglect, domestic violence, loss of a parent, and the ACE-DEP score. PC lifetime SBs were significantly and positively correlated with mental and physical trauma, mental neglect, domestic violence, and the ACE-DEP score, while PC current SBs were associated with the same items and additionally with sexual abuse and the loss of a parent. The phenome score was significantly correlated with the same items but additionally with a family history of mental disease.

### 3.5. Best Prediction of the Phenome

Table 4 shows the three different models which predict the phenome score. Regression #1 shows that 47.7% of the phenome score was explained by the ACE-DEP score (highly significant), while age, gender, and education were not significant. Figure 1 shows the partial regression of the phenome on the ACE-DEP scores. Regression #2 shows that after intruding ROI and the neuroimmunotoxic and CIRS profiles, ACE-DEP was no longer significant, indicating that the effects of ACE-DEP are mediated by ROI and the immune profiles. Regression #3 shows the best prediction, whereby 80.1% of the variance in the phenome is explained by ROI and neuroimmunotoxicity (both positively associated) and age and CIRS (both inversely associated). We have rerun this analysis with the other immune profiles entered instead of neuroimmunotoxicity and found that IRS (β = 0.74, t = 3.02, *p* = 0.004), M1 (β = 0.425, t = 3.69, *p* < 0.001), Th1 (β = 0.214, t = 2.63, *p* = 0.012), Th2 (β = 0.265, t = 2.72, *p* = 0.009), Th17 (β = 0.287, t = 2.96, *p* = 0.005), and T cell growth (β = 0.261, t = 2.05, *p* = 0.046) were significantly associated with the phenome.

Regression #4 shows that the ROI scores were best predicted by ACE-DEP, which explained 37.9% of the variance in the phenome. Figure 2 shows the partial regression of ROI on the ACE-DEP scores.

### 3.6. Associations between ACEs and Immune Profiles

Table 4 shows the correlations between the ACEs and the IRS, neuroimmunotoxic, and growth factor profiles. The latter were entered as the residualized profiles after partialling out the effects of the baseline levels (thus examining the association with the actual changes in production following stimulation). The residualized IRS, neuroimmunotoxic, and growth factor profiles were significantly associated with mental and physical trauma, mental neglect, and the ACE-DEP score. Moreover, a family of mental health issues was associated with the IRS scores. In Table 5, we examined the regressions of the immune profiles on ACE-DEP after allowing for the effects of age, sex, BMI, and TUD. The results show that around 12–22% of the variance in the immune profiles was explained by ACE-DEP independently of age, sex, BMI, and TUD. Moreover, entering the DSM-5 diagnosis of major depression, the HDRS score, and the drug state (sertraline, other antidepressants, mood stabilizers, antipsychotics, and benzodiazepines) in the automatic regression showed that ACEs-DEP, and not the diagnosis, severity of illness, or the drug state, was the significant explanatory variable. Figure 3 shows the partial regression of the growth factor profile on the ACE-DEP score.

### 3.7. Construction of Pathway Phenotypes and Results of PLS Analysis

The results of the PLS path analysis on 5.000 bootstrap samples after a feature and path selection are displayed in Figure 4. The phenome was conceptualized as an LV extracted from HDRS, STAI, LV recent SBs, and the phenome score (including the impact of depression with melancholia and psychotic features). This LV construct’s reliability was good, with a Cronbach’s alpha of 0.924, an rho A of 0.946, a composite reliability of 0.946, and an AVE of 0.816, and the outer model loadings were all larger than 0.84, with a *p*-value of <0.0001. We were able to construct a ROI-IMMUNE pathway phenotype comprising ROI features and the three key immune profiles. This LV also showed excellent psychometric properties with a Cronbach’s alpha of 0.869, an rho A of 0.898, a composite reliability of 0.897, and an average variance extracted (AVE) of 0.594, and the outer model loadings were all larger than 0.64 at *p* < 0.0001. ACE-DEP was conceptualized as an LV extracted from four ACEs and showed adequate quality criteria, namely a Cronbach’s alpha of 0.776, an rho A of 0.796, a composite reliability of 0.854, and an average variance extracted (AVE) of 0.595, and the outer model loadings were larger than 0.68 at *p* < 0.0001. The model’s overall fit was satisfactory with SRMR = 0.075. Blindfolding demonstrated that the cross-validated redundancies of the phenome (0.649) and ROI-IMMUNE (0.289) and the cross-validated communalities of ACE-DEP (0.332) were appropriate. We observed that 82.2 percent of the variance in the phenome LV was explained by the regression on ROI-IMMUNE LV, sexual abuse (both positively) and age and CIRS (both inversely). ACE-DEP explained 50.4% of the variance in the ROI-IMMUNE LV and 17.8% of the variance in CIRS. There were significant specific indirect effects of ACE-DEP on the phenome that were mediated by CIRS (*t* = −2.22, *p* = 0.026) and ROI-IMMUNE LV (t = 8.06, *p* = < 0.001), leading to a significant total effect of ACE-DEP (t = 7.70, *p* < 0.001). The ROI-IMMUNE LV explained 66.8% of the variance in the phenome, and the ROI-IMMUNE (positively) and CIRS (inversely) explained 73.7% of the variance in the phenome. PLSpredict shows that the Q2 Predict values for all the indicators of the endogenous constructs were positive, suggesting that they surpassed the naïve benchmark (the prediction error was less than the error of the naivest benchmark). Compositional invariance was shown by combining predicted–oriented segmentation analysis with multi-group analysis.

Exploratory factor analysis showed that one general factor could be extracted from the phenome (factor loading: 0.844), ROI (0.753), ACE-DEP (0.722), and the growth factor (0.692) and IRS (0.708) profiles, which explained 55.6% of the variance (KMO = 0.670, Bartlett’s χ2 = 181.542, df = 10, *p* < 0.001).

### 3.8. Results of Network, Annotation, and Enrichment Analysis

#### 3.8.1. All ACE DEPs

Figure 5 shows the first-order PPI network build around the upregulated DEPs of ACE (comprising 50 interactions in the first shell and none in the second shell, evidence level = 0.400). This PPI comprises 65 nodes with 829 edges, exceeding the predicted number (*n* = 179) with a PPI-enrichment value of *p* < 1 × 10^−16^. This network shows the following features: network diameter: 3, radius: 2, typical path length: 1.623, average number of neighbors: 25.5, clustering coefficient: 0.713, network density of 0.399, and a heterogeneity of 0.491. The top five seed hubs were TNF (degree = 53), VEGFA (46), CXCL8 (45), IL2 (44) and CSF3 (40). STAT3 (52) and FOXP3 (35) were the top non-seed genes in this network. The top two non-hub bottlenecks were FGF2 (betweenness centrality = 0.0200) and PDGFA (0.0103).

Figure 6 displays the results of GOnet enrichment analysis and the most important (q threshold: *p* < 0.0001, restricted graph *p* value < 1.2 × 10^−9^) GO annotations. ESF Appendix A shows the enriched ontology term clusters in the PPI network of ACE, indicating that cytokine signaling, chemotaxis, responsivity to an external stimulus, and cytomegalovirus are the major term clusters. ESF Appendix A displays the Voronoi diagram of the hierarchical Reactome pathways, showing that (apart from immune system and chemokine receptors) the most important terms were the diseases of signal transduction by growth factor receptors and second messengers, G protein-coupled receptors (GPCR), phosphatidylinositol 3-kinase (PI3K) cascade, receptor tyrosine kinases (STAT3), and mitogen-activated protein kinase (MAPK).

Table 6 summarizes the results of an enrichment analysis performed on the ACE PPI network using OmicsNet and IntAct. The intracellular protein kinase cascade, nuclear factor (NF)-κB, viral reproduction, and MAPK pathways were the top biological processes enriched in all DEPs. The top over-represented PANTHER biological processes were the viral, apoptotic, immune, and rhythmic and circadian processes.

Table 7 shows the top Kegg pathways that were overrepresented, namely the immune, viral (cytomegalovirus), IL-17, TNF, and Janus kinases/signal transducer and activator of transcription (JAK-STAT) pathways. Table 8 shows the results of Metascape MCODE analysis (GO biological functions and molecular functions, GO and CORUM cellular components, and Kegg pathways), which identified two molecular complexes: (a) response to cytokines and (b) TNFR/MAPK signaling pathways.

Table 9 shows the top 10 inBio Discover DOID terms that were associated with the ACE PPI network, including immune and autoimmune disorders. The same table also shows the results of the inBio Discover analysis using targeted custom DOID, WP, and GO terms revolving around brain disease, neuronal functions, and atherosclerosis.

#### 3.8.2. DEPs of the Growth Factor Cluster

Figure 5 shows that in using MCL cluster analysis (with an inflation parameter of 3) two clusters could be formed; the first was built around immune DEPs and the second around growth factor DEPs. Consequently, we examined the terms that are overrepresented in the growth factor network. Figure 7 shows a bar graph made using Enrichr and Appyter, indicating that the 10 GO biological processes which are overrepresented in the growth factor network are endothelial cell proliferation, regulation of phosphorylation, and the MAPK cascade.

Table 7 shows the top Kegg and GO pathways enriched in the enlarged growth factor network, namely Ras-associated protein 1 (Rap1)/Ras/MAPK and PI3K/protein kinase B (Akt)/mammalian target of rapamycin (PI3K-Akt-mTOR) signaling pathways and angiogenesis and endothelial cell proliferation. Table 8 shows that two molecular clusters were extracted from the growth factor DEP network, namely (a) cellular response to growth factors and (b) response to hypoxia.

ESF Appendix A shows a bar graph (made using Enrichr and Appyter) indicating that Rap1, Ras, calcium, MAPK, and P13K-Akt signaling are the major Kegg 2021 pathways. ESF Appendix A displays a bar graph (made using Enrichr and Appyter) indicating that Akt-mTOR signaling and angiogenesis are the top pathways in the growth factor PPI network.

## 4. Discussion

### 4.1. Activated Immune Profiles Due to ACEs

The first major finding of this study is that ACEs, including mental and physical trauma are accompanied by activation of different immune profiles, including M1 macrophage, Th1, Th2, and Th17 profiles and, as a consequence, IRS and neuroimmunotoxic and T cell and growth factor profiles. Increased ACE scores explain a larger part in the IRS (19.5%), neuroimmunotoxic (15.1%), and growth factor (18.1%) profiles than in the CIRS (10.1%) profile. Most importantly, the effects of ACE on these immune profiles were more prominent than the impact of the diagnosis of major depression or severity of illness, indicating that ACEs, and not the current mood state, determine those changes in immune profiles. The latter findings indicate that negative recall bias, which is associated with the current mood state [55], does not explain the ACE-immune relationship. Furthermore, our analyses disclosed that the stimulated (and not the unstimulated) production of 12 cytokines/chemokines, namely IL-2, IL-5, IL-9, IL-12, IL-15, IL-17, G-CSF, sIL-1RA, TNF-α, CXCL8, CXCL10, and CCL5, and 3 growth factors, namely FGF, PDGF, and VEGF, were elevated by ACEs.

In our study, mental trauma, physical trauma, mental neglect, domestic violence, a family history of mental illness, and losing a parent to separation, death, or divorce belonged to one and the same factor that was strongly associated with major depression, whilst sexual abuse did not belong to this factor but was associated with depression. Moreover, physical neglect, witnessing a mother being abused, and a family member with drug abuse were not associated with depression. Importantly, a score of ≥3 ACEs on the first factor was associated with enhanced immune responsivity, whereas subjects with a lower score, and those with other ACEs (including sexual abuse) did not show changes in the stimulated cytokine/growth factor production.

Overall, our results extend those of previous reports that ACEs are associated with selected immune biomarkers, including serum/plasma levels of IL-6, TNF-α, and soluble urokinase plasminogen activator receptor [36,40,42,56]. While previous reports could not establish an association between CRP and ACEs [56], Moraes et al. [12] detected a significant association between sexual abuse and CRP in women with BD. In addition, a previous report showed that sexual abuse was associated with lowered levels of antioxidant defenses, including zinc, albumin, and -SH groups [27]. Therefore, it appears that the type of ACE (our first ACE factor versus physical neglect and versus sexual trauma) may play a key role in the ACE’s associations with biomarkers. This is further underscored by our findings that physical neglect is not associated with the immune profiles measured here, whilst we previously detected that physical neglect was the major determinant of increased RONS/OSTOX [27]. Future research should examine the differential effects of these ACEs on immune versus nitro-oxidative pathways.

A noteworthy contrast between the present and prior studies is that the current study employed a culture supernatant of unstimulated and stimulated diluted whole blood to measure a panel of 27 cytokines/growth factors, whereas previous papers measured a few inflammatory biomarkers in serum/plasma. As such, we obtained immune measurements which reflect the baseline immune condition (unstimulated culture) versus a polyclonally stimulated immune profile which reflects responsivity to immune stimuli. Furthermore, the diluted whole blood technique used here accurately represents the in vivo cytokine/growth factor production following immune stimuli because the original cell-to-cell interactions are preserved in whole blood [49,50,51]. Moreover, our technique allows the measurement of the production of cytokines/growth factors which are hardly measurable in serum/plasma, including IL-2, IL-5, IL-9, IL-12, IL-15, IL-17, and VEGF and, therefore, allows a more precise measurement of both baseline and polyclonally stimulated immune profiles.

In this respect, it is important to note that there were no effects of ACEs on the unstimulated immune profiles, whereas all the PHA+LPS-stimulated immune profiles were strongly elevated by ACEs. Moreover, not the unstimulated production but the residualized (baseline levels partialled out) M1, Th1, Th2, Th17, IRS, neuroimmunotoxic, and growth factor production profiles were predicted by the ACE scores. Therefore, it is safe to conclude that the ACEs may sensitize key components of the immune system and that later immune triggers with similar properties to mitogens and LPS activate the sensitized cytokine/growth hormone responses, leading to elevated IRS responsivity. Phrased differently, interactions between ACE-sensitized immune profiles and new immune stimuli appear to activate the immune system, leading to IRS-associated neuroimmunotoxicity.

### 4.2. ACEs, ROI-IMMUNE Pathway Phenotype and the Phenome

The second major finding of this study is that the ACE score significantly predicted ROI and the affective phenome and that the effects of ACEs on the phenome were completely mediated by a newly constructed ROI-IMMUNE pathway phenotype (positively) comprising ROI features, IRS, neuroimmunotoxicity and growth factors. Previously, it was detected that ACEs predict the phenome of affective disorders [1,2] and that these effects are mediated by a ROI-REDOX pathway phenotype, conceptualized as a latent vector extracted from the ROI and nitro-oxidative pathways [2,3]. Based on these findings, the affective neuroimmunotoxicity theory of affective disorders was coined which conceptualizes that increased neurotoxicity due to immune-nitro-oxidative damage and lowered antioxidant defenses is associated with ROI, thereby causing ROI-associated recurrent damage to affective circuits in the brain [1,2]. Previously, we pointed out that MDD/MDE demonstrates heightened neuroimmunotoxicity due to increased production of IL-1β, IL-6, TNF-α, IL-17, IL-2, IFN-γ, CXCL8, CXCL10, and CCL5, all of which have neuroimmunotoxic characteristics [1,2,3,32,34]. Therefore, the results of the present study indicate that ACEs predispose enhanced neurotoxicity and, consequently, affective symptoms by activating IL-2, IL-12, IL-15, IL-17, TNF-α, CXCL8, CXCL10, and CCL5, which all have neurotoxic effects [32,34]. The neuroimmunotoxic effects of ACEs on affective symptoms may compound the neurotoxic implications of increased RONS/OSTOX and lowered antioxidant defenses [1,2,3,57].

Additionally, the current study’s findings indicate that ACEs stimulate the production of VEGF, PGDF, and FGF, whereas previous research indicated that increased FDF concentrations were associated with depression, whilst findings on VEGF and PDGF levels were more contentious [58,59,60,61,62,63]. Nonetheless, growth factors such as VEGF are sometimes difficult to quantify in serum [64] but are well quantifiable in diluted whole blood cultures (this study). We may infer from the above that the link between depression and growth factors may be explained by the effects of ACEs. This is important because the subnetwork of the growth factors measured here interacts with the cytokine network, thereby contributing to immune responsivity and immune activation via different pathways, as described in Section 4.3.

Our PLS analysis revealed that the ROI-IMMUNE pathway phenotype (positively) and CIRS (inversely) explained 73.7% of the variance in the affective phenome. These results confirm the IRS/CIRS hypothesis of depression, according to which elevated IRS (M1, Th1, Th17) coupled with attenuated CIRS (Th2 and Treg) profiles determine the phenome of acute episodes [32]. Nonetheless, our findings indicate that ACEs have a greater stimulatory effect on the IRS than the CIRS profiles, suggesting that IRS activities are not dampened by CIRS upon re-activation of the immune system, resulting in increased IRS and neuroimmunotoxic responses [32]. Additionally, it is critical to emphasize that, in addition to the effects of ACEs, which are mediated by the ROI-IMMUNE pathway phenotype, sexual abuse has an influence on the phenome and that this effect is not mediated by the immune pathways evaluated here.

In keeping with the new approach to precision nomothetic psychiatry [3,29], the current study developed novel precision constructs that included not only the ROI-IMMUNE pathway phenotype as discussed above, but also a replicable and validated factor derived from ACEs, ROI, immune profiles (e.g., growth factors and IRS), and the affective phenome. The first pathway phenotype demonstrates that ACEs account for about 50% of the variation in the ROI-IMMUNE pathway phenotype, implying that ROI and activated immune pathways are manifestations of a shared underlying construct that is highly impacted by ACEs. As such, ACEs seem to be associated with the recurrence of affective episodes and suicidal behaviors. In this regard, a recent meta-analysis showed that a lifetime history of suicide attempts is strongly related with the activated immune-inflammatory and O&NS pathways in affective disorders [65,66]. As a result, it is reasonable to argue that ACEs induce ROI, which is accompanied by immune sensitization which, upon new immune hits, may result in activated IRS and neuroimmunotoxic pathways and, consequently, the onset of a new episode.

The second pathway phenotype constructed here demonstrates that ACEs, ROI, increased immune responsiveness, and the affective phenome are all manifestations of a common core, namely the trajectory of affective disorders across distinct lifetime epochs, beginning with ACEs, sensitized immune responses, novel (immune) hits activating the sensitized immune system, and recurrent episodes of affective disorders and suicidal behaviors. Recently, a comparable ACE-based pathway phenotype was created, namely an ACE-ROI latent vector that was substantially related with nitro-oxidative neurotoxicity and the affective disorder phenome [1,2,3]. Overall, our findings indicate that the cumulative impact of ACEs, ROI, and immunological responses substantially predicts the phenome of an acute depressive episode, including current suicidal behaviors. According to a recent meta-analysis, current suicidal behaviors, including suicidal ideation, are related with active neurotoxic pathways mediated by the IRS and OSTOX pathways [65,66].

### 4.3. Network, Enrichment, and Annotation Analysis

The third major findings of this study are the results of network, annotation, and enrichment analysis showing which molecular functions and pathways are sensitized by ACEs. A first conclusion of this analysis is that the PPI network of ACEs exhibit a high degree of connectedness and two interrelated communities, one concentrated on immune DEPs and the other on growth factors. The network’s backbone is made up of DEPs that contribute to both communities, namely TNF, CXCL8, IL2, and CSF3 and VEGFA, FGF2, and PDGFA. Non-seed genes that are important hubs and bottlenecks are STAT3 and FOXP3. As a result, it looks as if ACEs induce an intertwined response in a network composed of highly coupled growth factors and immune clusters. In this respect, we found that these three growth factors influence cell division, the MAPK signaling pathways, and especially PI3K/Akt/mTOR and Rap1/Ras/MAPK signaling, which are the main proliferation/survival pathways [67]. As such, the ACE-induced sensitization of the growth factors contributes to the sensitization and, consequently, IRS activation and enhanced neuroimmunotoxic responses.

The top pathways and molecular functions that are over-represented in the PPI network of ACEs comprise inflammation and chemotaxis, the JAK-STAT pathway, including STAT3, NF-κB, and TNF/apoptotic, and GPCR signaling. The JAK-STAT, TNFR1-induced NF-κB signaling, and TNF-α/death receptor signaling are key pathways involved in IRS signaling [68,69,70,71,72]. These findings indicate that STAT3 and FOXP3 are predicted to be key factors associated with ACEs. The JAK-STAT pathway is involved in inflammation, T cell proliferation, cell division, and death, while STAT3 is associated with autoimmune reactions [68,69,70]. Furthermore, cytokines such as IL-2, IL-5, IL-9, IL-12, IL-15, and IFN-γ and GPCR and growth factors signal via the JAK-STAT pathway, thereby transactivating Janus kinases and resulting in the nuclear translocation of STATs and the upregulation of cytokine-modifiable genes [68]. Our enrichment analyses also discovered that ACEs are associated with the TNF-α, IκB kinase (IKK), and NF-κB cascade, whereby the latter serves as a transcriptional activator of the expression of various cytokine genes [73].

Moreover, other significant functions and paths enriched in the growth factor networks of ACEs are angiogenesis and endothelial cell proliferation and atherosclerosis. Such effects, coupled with the IRS response, may explain the association between ACEs and the development of atherosclerosis and ischemic heart disease in later life [74,75]. Our growth factor PPI network was highly significantly associated with a cellular response to hypoxia, and the PPI network comprised hypoxia-related genes, including the hypoxia-inducible factor 1A (HIF1A) gene. This is important because affective symptoms due to acute COVID-19 [76] and long COVID-19 (to be submitted) are largely the consequence of hypoxemia. Finally, the growth factor PPI network was enriched in rhythms and circadian rhythms. Many growth factors show a circadian variation, including FGF [77], which in turn regulates circadian behaviors as a feature of an adaptive starvation response [78]. VEGF is one of the CLOCK-controlled genes which may elicit downstream effects, including on angiogenesis, period, and cryptochrome family members [79]. Cryptochrome is expressed in the central nervous system and mediates behavioral avoidance responses [80]. Moreover, the CLOCK-controlled genes are regulated by STAT-3 and probably HIF1A [81], which belong to the ACE PPI network.

Finally, our enrichment analyses also disclosed that the cytokine/growth factor profile of ACEs is associated with many immune and autoimmune disorders, such as arthritis, inflammatory bowed disease, demyelinating and neuroinflammatory disease, and atherosclerosis, which show a strong comorbidity with MDD/MDE, which was previously ascribed to the activated IRS and OSTOX pathways [82]. Importantly, biological process analyses revealed that the ACE PPI network is associated with a cellular response to a bacterium and LPS, as well as viral infections, including cytomegalovirus. This may indicate that an increased LPS load, due, for example, to the translocation of commensal bacteria following leaky gut [83], may be one of the trigger factors that, coupled with sensitized immune pathways, lead to a new episode. Previously, we reported that anti-human cytomegalovirus IgG levels interact with BD to attenuate the expression of the CIRS T cell phenotype CD4+CD25+FOXP+GARP [84]. As such, latent cytomegalovirus infections could interfere with CIRS functions, thereby increasing the propensity towards IRS and neuroimmunotoxic responses.

## 5. Limitations

The current study’s findings should be discussed in the light of its limitations. First, this study would have been more interesting if we also had measured biomarkers of oxidative and nitrosative stress, as well as other growth factors and inflammatory mediators. Second, although well-powered, the study was conducted on a smaller sample of 20 healthy controls and 30 depressed patients.

## 6. Conclusions

Figure 8 summarizes the findings of the present study. The cumulative effects of mental and physical trauma, mental neglect, domestic violence, a family history of mental disease, and the loss of a parent resulted in increased stimulated production of M1, Th1, Th2, Th17, IRS, neuroimmunotoxicity, and GF profiles and predicted a significant portion of the variance in ROI and the phenome of mood disorders. We constructed a new pathway phenotype by combining ROI features (number of episodes and lifetime suicidal attempts and suicidal ideation) with IRS/neuroimmunotoxic/growth factor profiles. PLS pathway analysis revealed that the combined impacts of this ROI-IMMUNE pathway phenotype (positively) and CIRS (inversely) explained a major portion of the variance in the phenome. Moreover, the effects of ACEs on the phenome are completely mediated by the ROI-IMMUNE pathway phenotype. Furthermore, we also constructed a second pathway phenotype as a latent vector extracted from ACEs—ROI—immune responsiveness—the affective phenome, indicating that these four indicators are manifestations of a common core, namely the trajectory of affective disorders across distinct lifetime epochs, beginning with ACEs, sensitized immune responses, novel (immune) hits activating the sensitized immune system, and recurrent episodes and suicidal behaviors. The enrichment analysis revealed that ACE-associated sensitization of the immune/GF profiles may be explained by the JAK-STAT pathway, NF-κB, TNF, and GPCR pro-inflammatory signaling, as well as hypoxia, angiogenesis, and the/Akt/RAS/MAPK pathways. The latter is the main proliferation/survival pathway, which is sensitized by ACEs and upon renewed activation may further boost the IRS and neuroimmunotoxic pathways. The immune profile of ACEs predicts that ACEs may increase the vulnerability to the development of many immune-inflammatory and autoimmune disorders. Flare-ups of the latter and viral and bacterial infections may consequently activate the sensitized immune/growth factor profiles causing the onset of new affective episodes. Moreover, we previously found that physical neglect and sexual abuse impacted nitro-oxidative and antioxidant pathways, which contribute to the phenome of mood disorders. The ACE-induced immune/growth factor responses, the backbone of the PPI network, and the molecular pathways underpinning these responses are new possible drug targets in the treatment of ACE-associated depression.

## Figures and Tables

**Figure 1 cells-11-01564-f001:**
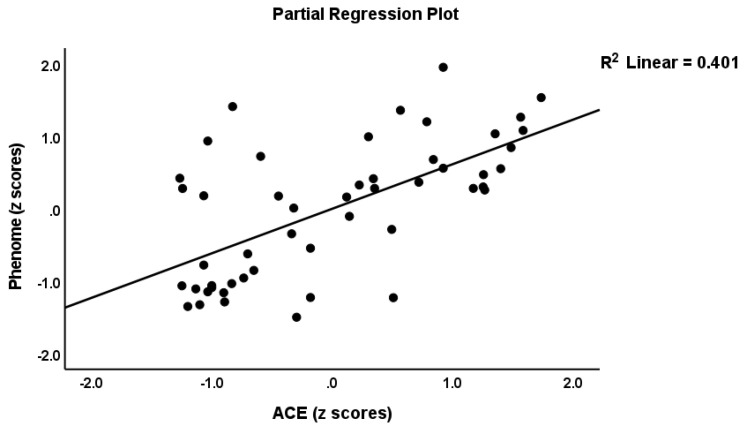
Partial regression of the phenome score on the Adverse Childhood Experience (ACE) score.

**Figure 2 cells-11-01564-f002:**
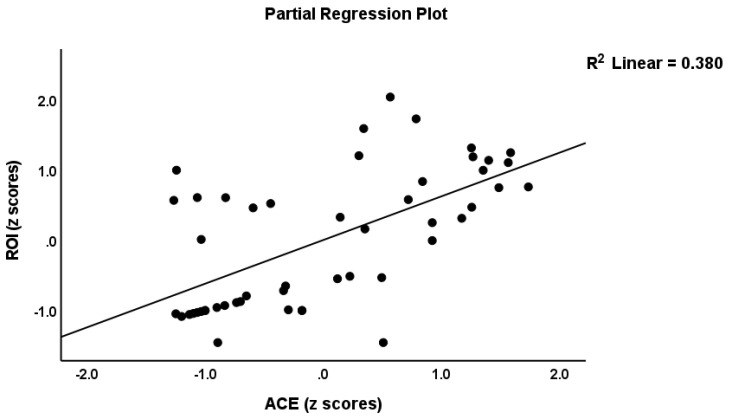
Partial regression of the reoccurrence of illness (ROI) index on the Adverse Childhood Experiences (ACE) score.

**Figure 3 cells-11-01564-f003:**
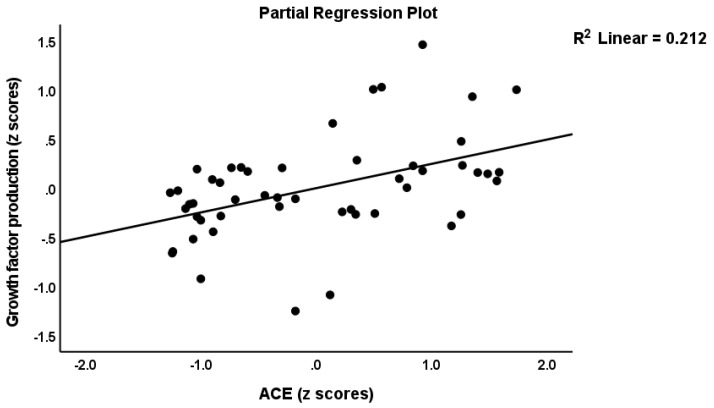
Partial regression of the growth factor profile on the adverse childhood experiences (ACE) score.

**Figure 4 cells-11-01564-f004:**
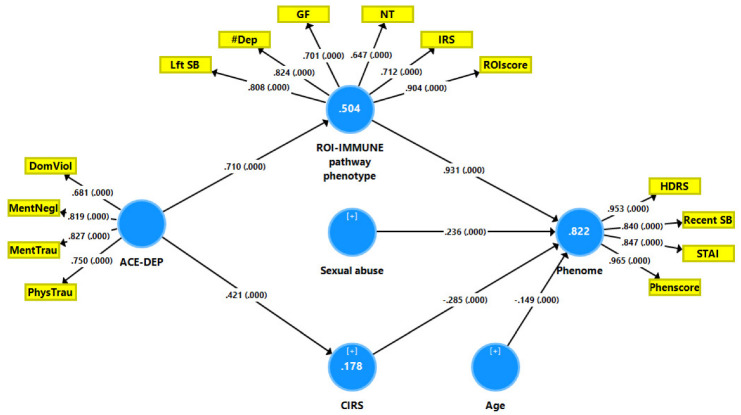
Results of Partial Least Squares analysis with the phenome of depression as the outcome variable and the effects of adverse childhood events (ACEs) on the phenome being mediated by the recurrence of illness (ROI) and immune biomarkers. The phenome of depression is entered as a latent vector (LV) extracted from the HDRS (Hamilton Depression Rating Scale) and STAI (State-Trait Anxiety Inventory) scores, recent suicidal behaviors (SB), and the phenome score (Phenscore), including melancholia and psychosis. ACE was conceptualized as an LV extracted from 4 ACEs, namely domestic violence (DomViol), mental neglect (MentNegl), and mental (MentTrau) and physical (PhysTrau) trauma. ROI-IMMUNE: a common core extracted from ROI features and immune profiles, i.e., Lifetime (Lft) SB, number of lifetime depressions (#Dep), ROI score, immune-inflammatory response (IRS), neuroimmunotoxicity (NT), and the growth factor (GF) immune profiles. CIRS: compensatory immunoregulatory profile.

**Figure 5 cells-11-01564-f005:**
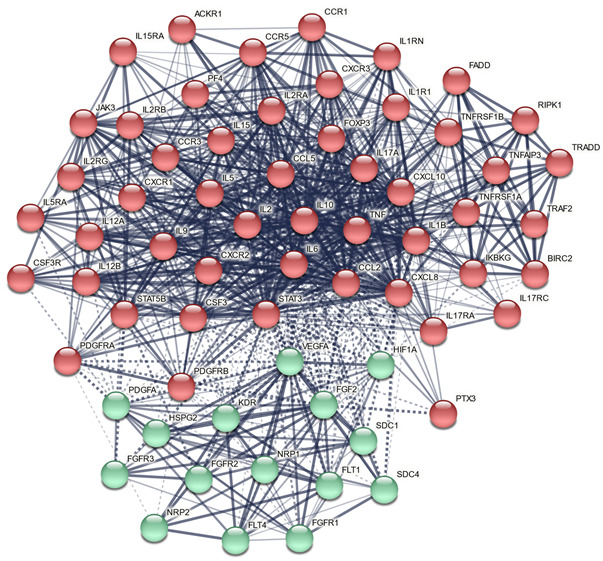
First-order protein-protein interaction (PPI) network built around the differentially expressed proteins of Adverse Childhood Experiences with the results of Markov clustering analysis. The solid and dotted lines represent connections inside and between clusters, respectively. Red: immune cluster; green: growth factor cluster.

**Figure 6 cells-11-01564-f006:**
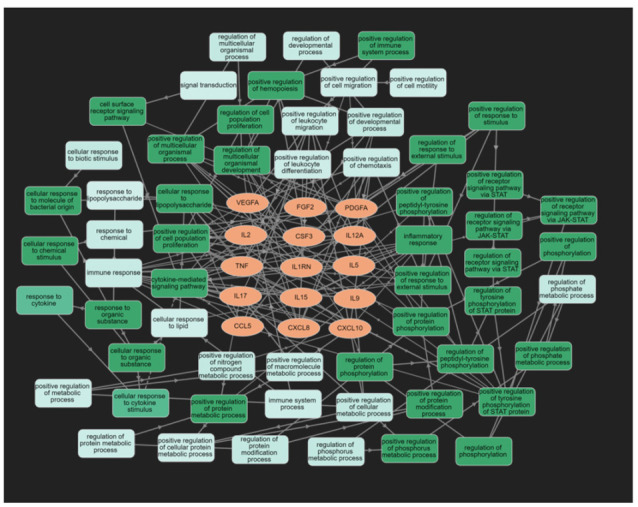
Results of GOnet enrichment analysis showing the 15 upregulated differentially expressed proteins of Adverse Childhood Experiences and their significant GO annotations.

**Figure 7 cells-11-01564-f007:**
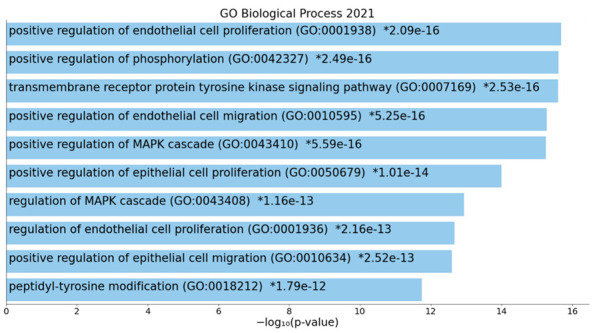
Bar chart with the top ten GO biological processes which were overrepresented in the protein-protein interaction network of Adverse Childhood Experiences.

**Figure 8 cells-11-01564-f008:**
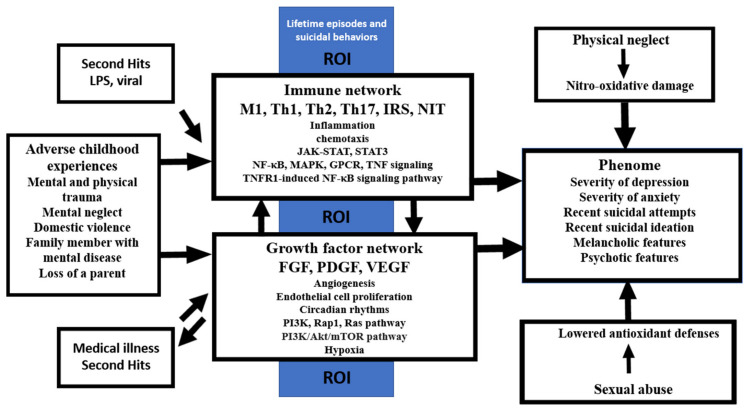
Summary of the findings of the current study. ROI: reoccurrence of illness index (ROI); M1 macrophage; Th: T helper; IRS: immune-inflammatory responses system; NIT: neuroimmunotoxicity; JAK-STAT: Janus kinases/signal transducer and activator of transcription; NF: nuclear factor; MAPK: mitogen-activated protein kinase; GPCR: G protein-coupled receptors; TNFR: tumor necrosis factor receptor; FGF: fibroblast growth factor; PDGF: platelet-derived growth factor; VEGF: vascular endothelial growth factor; Rap1: Ras-associated protein 1; PI3K-Akt-mTOR: phosphatidylinositol 3 kinase/protein kinase B/mammalian target of rapamycin.

**Table 1 cells-11-01564-t001:** Demographic and clinical data, including adverse childhood experiences in healthy controls (HC) and patients with major depression.

Variables	HC (*n* = 20)	Major Depression (*n* = 30)	F/X^2^/FEPT	df	*p*
Sex (Male/Female)	6/14	11/19	0.24	1	0.626
Age (years)	33.6 (8.0)	28.7 (8.6)	2.47	2/47	0.095
Education (years)	16.1 (2.2)	15.6 (1.4)	2.99	2/47	0.060
BMI (kg/m^2^)	21.33 (2.51)	25.52 (5.95)	4.32	2/47	**0.0** **19**
TUD (No/Yes)	18/2	23/7	FEPT	-	0.285
HDRS	0.9 (1.5)	23.5 (5.8)	147.01	2/47	**<0.001**
STAI	37.8 (10.6)	56.8 (7.2)	28.00	2/47	**<0.001**
Total number of all episodes	0.0	2.10 (0.92			
Reoccurrence of illness	−1.089 (0.00)	0.726 (0.586)	KW	-	**<0.001**
Lifetime suicidal behaviors	−0.987 (0.0)	0.658 (0.767)	KW	-	**<0.001**
Recent suicidal behaviors	−0.916 (0.0) ^c^	0.611 (0.861)	KW	-	**<0.001**
LV phenome	−1.123 (0.225)	0.749 (0.455)	170.48	2/47	**<0.001**
Mental trauma	19/1	13/17	13.90	1	**<0.001**
Physical trauma	19/1	16/14	9.92	1	**0.002**
Sexual abuse	20/0	22/8	FEBT		**0.015**
Mental neglect	20/0	14/16	15.69	1	**<0.001**
Physical neglect	17/3	27/3	FEPT	-	0.672
Domestic violence	119/1	19/11	FEPT	-	**0.016**
Drug abuse in family	19/1	29/1	FEPT	-	1.00
Family history of mental illness	20/0	18/12	FEPT	-	**0.001**
Losing a parent	18/2	19/11	4.44	1	**0.035**
Criminal behavior	18/2	29/1	FEPT	-	0.556
ACE domain 1	0.25 (0.55)	2.70 (1.82)	39.66	1/48	**<0.001**
ACE domain 2	0.300 (0.66)	0.43 (0.68)	0.48	1/48	0.494

Results are shown as mean ±SD. F: results of analysis of variance; X^2^: analysis of contingency tables; FEPT: Fisher Exact probability test; LV: latent vectors; BMI: body mass index; TUD: tobacco use disorder; HDRS: Hamilton Depression Rating Scale score; STAI: Spielberger State and Trait Anxiety, State version; ACE: adverse childhood event; ACE domain 1 comprises mental trauma, physical trauma, mental neglect, domestic violence, family history of mental disease, and loss of a parent; ACE domain 2 comprises physical neglect, divorce, and criminal behaviors. Significant *p* values are shown in bold.

**Table 2 cells-11-01564-t002:** Differences in unstimulated (UNST) and lipopolysaccharide + phytohemagglutinin-stimulated (STIM) changes in various immune profiles in healthy controls (HC) and patients divided into those with high adverse childhood experiences (ACE ≥ 3) versus those with lower (ACE < 3) ACE scores.

Variables (z Scores)		HC ^a^ *n* = 20	ACE < 3 ^b^ *n* = 11	ACE ≥ 3 ^c^ *n* = 19	Wald df = 2	*p*
M1	UNST	−0.879 (0.061)	−0.867 (0.068)	−0.837 (0.060)	7.80	**0.020**
	STIM	0.607 (0.043) ^c^	0.762 (0.132)	1.269 (0.227) ^a^		
Th1	UNST	−1.385 (0.074)	−1.549 (0.086)	−1.484 (0.058)	8.05	**0.018**
	STIM	0.222 (0.085) ^c^	0.284 (0.152)	0.776 (0.237) ^a^		
Th17	UNST	−1.672 (0.058)	−1.693 (0.043)	−1.743 (0.004)	6.74	**0.039**
	STIM	0.266 (0.073) ^c^	0.370 (0.103)	0.738 (0.196) ^a^		
Th2	UNST	−1.324 (0.074)	−1.345 (0.617)	−1.299 (0.084)	12.14	**0.002**
	STIM	0.061 (0.089) ^c^	0.304 (0.198)	0.902 (0.269) ^a^		
IRS	UNST	−1.521 (0.095)	−1.566 (0.110)	−1.496 (0.096)	12.65	**0.002**
	STIM	0.123 (0.049) ^c^	0.309 (0.160) ^c^	0.885 (0.234) ^a^		
CIRS	UNST	−0.924 (0.060)	−0.918 (0.067)	−0.787 (0.091)	5.07	0.079
	STIM	0.664 (0.083)	0.807 (0.139)	1.210 (0.175)		
Tcell	UNST	−1.471 (0.092)	−1.518 (0.119)	−1.370 (0.146)	13.73	**0.001**
	STIM	0.032 (0.048) ^c^	0.194 (0.175) ^c^	0.846 (0.242) ^a^		
GF	UNST	−0.849 (0.098)	−0.828 (0.132)	−0.649 (0.149)	13.88	**0.003**
	STIM	0.474 (0.014) ^c^	0.717 (0.172)	1.213 (0.235) ^a^		
NT	UNST	−1.615 (0.102)	−1.682 (0.117)	−1.687 (0.063)	9.15	**0.010**
	STIM	0.266 (0.065) ^c^	0.367 (0.117)	0.799 (0.197) ^a^		

Results of GEE analyses with immune profiles as dependent variables and time, group (depression versus controls), and time by group interactions as explanatory variables and age, sex, body mass index, and tobacco use as covariates. Shown are the time x group effects (Wald) with ^a, b, c^ indicating pairwise comparisons among the sample means; df: degrees of freedom; UNST: unstimulated whole blood cultures; STIM: stimulated whole blood cultures. All data are shown as estimated marginal means (mean ±SE). See ESF Appendix A for explanation of the profiles and cytokines measured in this study. M1: M1 macrophage; Th: T helper; IRS: immune-inflammatory response system; CIRS: compensatory immunoregulatory response system; Tcell: T cell growth; GF: growth factors; NT: neuroimmunotoxicity. Significant *p* values are shown in bold.

**Table 3 cells-11-01564-t003:** Differences in lipopolysaccharide + phytohemagglutinin-stimulated changes in cytokines/growth factors in healthy controls (HC) and patients divided into those with high adverse childhood experiences (ACE ≥ 3) versus those with lower (ACE < 3) scores.

Variables (z Scores)	HC ^a^	ACE < 3 ^b^	ACE > 3 ^c^	Wald (df = 2)	*p* Value
sIL-1RA	−0.272 (0.041) ^c^	−0.053 (0.125)	0.386 (0.229) ^a^	17.17 (G)	**<0.001**
IL-2	−0.222 (0.113)	0.088 (0.131) ^c^	0.653 (0.200) ^a^	6.34 (T x G)	**0.042**
IL-5	−0.303 (0.135) ^c^	0.053 (0.246)	0.764 (0.361) ^a^	11.24 (T X G)	**0.005**
CXCL8	−0.155 (0.018) ^c^	0.294 (0.357)	0.900 (0.392) ^a^	9.20 (T X G)	**0.010**
IL-9	−0.268 (0.034) ^c^	0.116 (0.281)	0.812 (0.358) ^a^	9.80 (T X G)	**0.007**
IL-12	−0.450 (0.031) ^c^	−0.325 0.102	−0.002 0.164 ^a^	7.29 (G)	**0.026**
IL-15	0.023 (0.029) ^c^	0.236 (0.191)	0.728 (0.229) ^a^	14.34 (T X G)	**<0.001**
IL-17	0.007 (0.063) ^c^	0.187 (0.180)	0.731 (0.230) ^a^	10.49 (T X G)	**0.005**
FGF	−0.757 (0.025) ^c^	−0.746 (0.044) ^c^	−0.577 (0.044) ^a, b^	14.45 (G)	**0.006**
G-CSF	−0.256 (0.014) ^c^	0.106 (0.262)	0.637 (0.354) ^a^	11.70 (T X G)	**0.003**
CXCL10	−0.809 (0.086) ^c^	-0.659 (0.161)	-0.336 (0.158) ^a^	6.94 (G)	**0.031**
MIP1A	−0.621 (0.085) ^c^	−0.635 (0.107) ^c^	−0.857 (0.031) ^a, b^	9.94 (G)	**0.007**
PDGF	−0.383(0.007) ^c^	−0.043 (0.228)	0.637 (0.349) ^a^	10.84 (T X G)	**0.004**
CCL5	−0.023 (0.132) ^c^	0.229 (0.232)	0.805 (0.303) ^a^	7.91 (T X G)	**0.019**
TNF-α	−0.283 (0.014) ^c^	−0.041 (0.180)	0.689 (0.397) ^a^	7.78 (T X G)	**0.020**
VEGF	0.050 (0.132) ^c^	0.292 (0.167)	0.734 (0.211) ^a^	8.03 (T X G)	**0.018**

Results of GEE analyses with cytokines/growth factors as dependent variables and time, group (depression versus controls), and time by group interactions as explanatory variables. Shown are the time x group effects (Wald) with ^a, b, c^ indicating pairwise comparisons among the groups All data are shown as estimated marginal means (mean ± SE) after covarying for age, sex, smoking, and body mass index. Significant *p* values are shown in bold.

**Table 4 cells-11-01564-t004:** Associations between 10 adverse childhood experiences and the reoccurrence of illness (ROI) index, lifetime and recent suicidal behaviors (SBs), the phenome of affective disorders, and immune and growth factor (GF) profiles.

Variables	ROI	Lifetime SB	Recent SB	Phenome	IRS	NIT	GF
Mental trauma	0.466 ***	0.388 **	0.342 *	0.547 ***	0.467 ***	0.401 **	0.447 ***
Physical trauma	0.422 **	0.432 **	0.319 *	0.455 ***	0.436 **	0.410 **	0.449 ***
Sexual abuse	0.231	0.225	0.315 *	0.347 *	−0.128	−0.186	−0.078
Mental neglect	0.646 ***	0.569 ***	0.574 ***	0.609 ***	0.464 ***	0.462 ***	0.403 **
Physical neglect	0.001	−0.043	0.035	−0.120	−0.171	−0.251	−0.011
Domestic violence	0.369 **	0.394 **	0.407 **	0.321 *	0.197	0.140	0.215
Drug abuse family	−0.108	−0.084	−0.024	−0.082	−0.040	−0.082	0.047
Family history	0.266	0.240	0.275	0.469 ***	0.280 *	0.261	0.266
Losing a parent	0.293 *	0.263	0.313 *	0.281 *	0.019	0.001	−0.022
Criminal	−0.099	−0.114	−0.034	−0.125	−0.159	−0.130	−0.189
ACE Domain1	0.633 ***	0.589 ***	0.582 ***	0.294 ***	0.413 **	0.337 *	0.473 ***

* *p* < 0.05; ** *p* < 0.01; *** *p* < 0.001. IRS: immune-inflammatory response system; NIT: neuroimmunotoxic; GF: growth factor profile.

**Table 5 cells-11-01564-t005:** Results of multiple regression analyses with the phenome of affective disorders, the reoccurrence of illness (ROI) index, or immune profiles as dependent variables.

Dependent Variables	Explanatory Variables	β	t	*p*	F _model_	df	*p*	R^2^
**1. Phenome**	**Model**				10.26	4/45	**<0.001**	0.477
	ACEs	0.622	5.48	**<0.001**				
	Age	−0.185	−1.90	0.118				
	Gender	−0.112	−0.99	0.325				
	Education	−0.021	−0.19	0.849				
**2. Phenome**	**Model**				37.21	5/44	**<0.001**	0.809
	ACEs	0.120	1.34	0.187				
	ROI	0.711	8.40	**<0.001**				
	Neuroimmunotoxicity	0.351	3.08	**0.004**				
	Age	−0.162	−2.37	**0.022**				
	CIRS	−0.227	−2.03	**0.048**				
**3. Phenome**	**Model**				45.26	4/45	**<0.001**	0.801
	ROI	0.775	11.05	**<0.001**				
	Neuroimmunotoxicity	0.388	3.47	**0.001**				
	Age	−0.181	−2.67	**0.010**				
	CIRS	−0.240	−2.13	**0.038**				
**4. ROI**	**Model**				29.31	1/48	**<0.001**	0.379
	ACEs	0.616	5.41	**<0.001**				
**5.1 M1 macrophage**	ACEs	0.384	2.88	**0.006**	8.30	1/48	**0.006**	0.147
**5.2 Thelper (Th)1**	ACEs	0.357	2.65	**0.011**	7.03	1/48	**0.011**	0.128
**5.3 Th2**	ACEs	0.475	3.74	**<0.001**	13.99	1/48	**<0.001**	0.226
**5.4 Th17**	ACEs	0.299	2.17	**0.035**	4.73	1/48	**0.035**	0.090
**5.5 IRS**	ACEs	0.442	3.22	**0.002**	11.62	1/48	**0.001**	0.195
**5.6 CIRS**	ACEs	0.317	2.32	**0.025**	5.37	1/48	**0.025**	0.101
**5.7 Neuroimmunotoxic**	ACEs	0.388	2.92	**0.005**	8.51	1/48	**0.005**	0.151
**5.8 T cell growth**	ACEs	0.452	3.51	**<0.001**	12.33	1/48	**<0.001**	0.204
**5.9 Growth factors**	ACEs	0.425	3.25	**0.002**	10.58	1/48	**0.002**	0.181

Phenome: conceptualized as the first principal component (PC) extracted from all symptom domains; ROI: computed as the first PC extracted from number of depressive and total episodes, lifetime number of suicidal attempts, and lifetime suicidal ideation; ACE: adverse childhood experiences computed as sum of mental and physical trauma, mental neglect, domestic violence, family history of mental disease, and loss of a parent; IRS: immune-inflammatory response system; CIRS: compensatory immunoregulatory system. Significant *p* values are shown in bold.

**Table 6 cells-11-01564-t006:** Top GO and PANTHER biological processes statistically overrepresented in the adverse childhood experiences network (analyzed with OmicsNet and Intact).

GO Biological Process	Total	Expected	Hits	*p*	pFDR
intracellular protein kinase cascade	1140	133	285	3.37 × 10^−39^	2.76 × 10^−36^
regulation of I−kappaB kinase/NF−kappaB cascade	210	24.7	90	1.77 × 10^−30^	7.25 × 10^−28^
I−kappaB kinase/NF−kappaB cascade	246	28.9	98	5.50 × 10^−30^	1.50 × 10^−27^
viral reproductive process	597	70.1	169	1.91 × 10^−29^	3.92 × 10^−27^
positive regulation of signal transduction	998	117	233	7.30 × 10^−27^	1.20 × 10^−24^
interaction with host	426	50	131	1.21 × 10^−26^	1.66 × 10^−24^
regulation of cellular protein metabolic process	1560	183	319	4.22 × 10^−26^	4.95 × 10^−24^
regulation of protein modification process	1250	147	266	3.03 × 10^−24^	3.11 × 10^−22^
positive regulation of I−kappaB kinase/NF−kappaB cascade	150	17.6	66	1.93 × 10^−23^	1.76 × 10^−21^
positive regulation of cellular protein metabolic process	968	114	218	6.06 × 10^−23^	4.97 ×10^−21^
positive regulation of protein modification process	867	102	201	9.54 × 10^−23^	7.11 ×10^−21^
regulation of protein metabolic process	1820	214	347	1.81 ×10^−22^	1.18 ×10^−20^
positive regulation of response to stimulus	1550	182	307	1.86 ×10^−22^	1.18 ×10^−20^
viral reproduction	803	94.3	189	3.39 ×10^−22^	1.90 ×10^−20^
regulation of MAPK cascade	559	65.6	147	3.48 ×10^−22^	1.90 ×10^−20^
**PANTHER Biological Process**	**Total**	**Expected**	**Hits**	** *P* **	**pFDR**
Viral process	448	52	150	6.09 × 10^−36^	1.18 × 10^−33^
Negative regulation of apoptotic process	577	67	160	1.34 × 10^−27^	1.30 × 10^−25^
Apoptotic process	699	81.2	163	1.84 × 10^−19^	1.19 × 10^−17^
Protein phosphorylation	627	72.8	140	3.93 × 10^−15^	1.91 × 10^−13^
Immune response	387	44.9	96	1.61 × 10^−13^	6.24 × 10^−12^
Rhythmic process	124	14.4	42	4.77 ×10^−11^	1.54 × 10^−09^
Angiogenesis	252	29.3	61	1.27 ×10^−08^	3.53 × 10^−07^
Cell−cell signaling	232	26.9	57	2.18 ×10^−08^	5.28 × 10^−07^
Circadian rhythm	90	10.5	27	2.07 ×10^−06^	4.47 ×10^−05^
Cell cycle	647	75.1	111	1.16 ×10^−05^	0.000224
Translation	315	36.6	62	1.85 ×10^−05^	0.000326
Cell proliferation	386	44.8	72	2.73 ×10^−05^	0.000441
Protein folding	157	18.2	36	4.21 ×10^−05^	0.000628
Protein folding	157	17.8	35	5.72 ×10^−05^	0.000925

FDR: False Discovery Rate.

**Table 7 cells-11-01564-t007:** KEGG pathway classifications of the differently expressed proteins of adverse childhood experiences (results of enrichment analysis using STRING).

Term ID all DEPs	Kegg Pathways	Observed	Background	Strength	FDR
hsa04060	Cytokine−cytokine receptor interaction	34	282	1.56	1.29 × 10^−41^
hsa04061	Viral protein interaction with cytokine and cytokine receptor	20	96	1.8	1.84 × 10^−27^
hsa05200	Pathways in cancer	29	517	1.23	4.06 × 10^−26^
hsa04630	JAK−STAT signaling pathway	20	160	1.58	9.98 × 10^−24^
hsa05163	Human cytomegalovirus infection	20	218	1.44	2.47 × 10^−21^
hsa04657	IL−17 signaling pathway	16	92	1.72	6.43 × 10^−21^
hsa04668	TNF signaling pathway	16	112	1.63	9.65 × 10^−20^
hsa04151	−Akt signaling pathway	20	350	1.24	1.05 × 10^−17^
hsa04659	Th17 cell differentiation	14	101	1.62	4.40 × 10^−17^
hsa05162	Measles	15	138	1.51	5.89 × 10^−17^
hsa04010	MAPK signaling pathway	18	288	1.27	1.41 × 10^−16^
hsa04061	Viral protein interaction with cytokine and cytokine receptor	22	96	1.84	2.86 × 10^−31^
**Term ID Cluster 2**	**Kegg Pathways**	**Observed**	**Background**	**Strength**	**FDR**
hsa04015	Rap1 signaling pathway	9	202	1.76	2.63 × 10^−12^
hsa04014	Ras signaling pathway	9	226	1.72	3.50 × 10^−12^
hsa04010	MAPK signaling pathway	9	288	1.61	1.95 × 10^−11^
hsa05205	Proteoglycans in cancer	8	196	1.73	6.14 × 10^−11^
hsa04151	−Akt signaling pathway	9	350	1.53	6.47 × 10^−11^
hsa01521	EGFR tyrosine kinase inhibitor resistance	6	78	2	1.41 × 10^−09^
hsa05200	Pathways in cancer	8	517	1.31	6.63 × 10^−08^
hsa05418	Fluid shear stress and atherosclerosis	5	130	1.7	1.73 × 10^−06^
hsa05230	Central carbon metabolism in cancer	4	69	1.88	8.78 × 10^−06^
hsa04510	Focal adhesion	5	198	1.52	1.06 × 10^−05^
hsa04810	Regulation of actin cytoskeleton	5	209	1.49	1.25 × 10^−05^
**Term ID Cluster 2**	**GO Biological Processes**	**Observed**	**Background**	**Strength**	**FDR**
GO:0001525	Angiogenesis	12	315	1.7	2.15 × 10^−15^
GO:0001936	Regulation of endothelial cell proliferation	9	134	1.94	4.05 × 10^−13^
GO:0001938	Positive regulation of endothelial cell proliferation	8	94	2.05	3.65 × 10^−12^
GO:0010595	Positive regulation of endothelial cell migration	8	103	2.01	6.16 × 10^−12^
GO:0050679	Positive regulation of epithelial cell proliferation	9	192	1.79	6.16 × 10^−12^
GO:0007169	Transmembrane receptor protein tyrosine kinase signaling pathway	11	518	1.44	6.18 × 10^−12^
GO:0050678	Regulation of epithelial cell proliferation	10	339	1.59	7.25 × 10^−12^
GO:0038084	Vascular endothelial growth factor signaling pathway	6	20	2.59	1.25 × 10^−11^
GO:0008284	Positive regulation of cell population proliferation	12	919	1.23	3.76 × 10^−11^
GO:0071363	Cellular response to growth factor stimulus	10	494	1.42	1.75 × 10^−10^

KEGG: Kyoto Encyclopedia of Genes and Genomes; ID: Identification; FDR: False Discovery Rate; DEP: differentially expressed proteins.

**Table 8 cells-11-01564-t008:** Results of Molecular Complex Detection (MCODE) analysis performed on all differentially expressed proteins (DEPs) and the growth factor cluster of adverse childhood experiences.

MCODE Components	ID	Annotations	Log10 (*p*) Value
All DEPs, MCODE1	Hsa 04060	Cytokine-cytokine interaction	−55.7
	GO:0071345	Cytokine-mediated signaling pathway	−50.7
	GO:0071345	Cellular response to cytokine stimulus	−42.8
All DEPs, MCODE2	CORUM:5531	Tumor necrosis factor receptor 1 signaling complex	−11.0
	Hsa 04010	MAPK signaling pathway	−10.0
	CORUM:6347	TNF-R1 signaling complex	−9.6
DEPs cluster 2, MCODE1	GO:0071363	Cellular response to growth factor stimulus	−39.8
	GO:0070848	Response to growth factor	−39.1
	GO:0007167	Enzyme-linked receptor protein signaling pathway	−36.7
DEPs cluster 2, MCODE2	GO:0001666	Response to hypoxia	−11.8
	GO:0036293	Response to decreased oxygen levels	−11.7
	GO:0071456	Cellular response to hypoxia	−11.6

Analyses are performed on all DEPs or on the growth factor DEPs (cluster2); ID: Identification; MAPK: mitogen-activated protein kinase; TNF-R: tumor necrosis factor receptor.

**Table 9 cells-11-01564-t009:** Results of inBio Discover annotation analysis with disease ontology annotations of diseases (DOID) performed on the upregulated differentially expressed proteins (DEPs) of adverse childhood experiences.

Term ID all DEPs	DOID Annotations	Size	Overlap	Enrichment	FDR
DOID:0060032	Autoimmune disease of musculoskeletal system	645	91/342	8.25	1.4 × 10^−57^
DOID:848	Arthritis	481	79/342	9.60	7.9 × 10^−55^
DOID: 3342	Bone inflammation disease	501	80/342	9.34	1.5 × 10^−54^
DOID:2914	Immune system disease	1.9 k	139/342	4.28	4.5 × 10^−54^
DOID:0050589	Inflammatory bowel disease	306	66/342	12.61	1.3 × 10^−53^
DOID:7148	Rheumatoid arthritis	313	65/342	12.14	1.1 × 10^−51^
DOID:612	Primary immunodeficiency disease	1.3 k	115/342	5.06	7.4 × 10^−51^
DOID:417	Autoimmune disease	1.1 k	104/342	5.70	1.9 × 10^−50^
DOID:65	Connective tissue disease	1.8 k	128/342	4.06	1.6 × 10^−46^
DOID:8893	Psoriasis	189	50/342	15.47	2.5 × 10^−45^
**Term ID all DEPs**	**Custom Term Annotations**	**Size**	**Overlap**	**Enrichment**	**FDR *p***
DOID: 3213	Demyelinating disease	218	42/342	11.27	5.8 × 10^−32^
DOID:1936	Atherosclerosis	352	34/342	5.65	3.1 × 10^−16^
DOID: 936	Brain Disease	1.5 k	59/342	2.29	1.5 × 10^−9^
WP 1455	Serotonin transported activity	11	3/342	15.95	7.4 × 10^−4^
GO:1901215	Negative regulation of neuron death	65	5/342	4.50	5.1 × 10^−3^
GO:1903978	Regulation of microglial cell activation	6	2/342	19.49	4.2 × 10^−3^

Term ID: term identification; FDR: false discovery rate.

## Data Availability

The dataset generated during and/or analyzed during the current study will be available from the corresponding author (M.M.) upon reasonable request and once the dataset has been fully exploited by the authors.

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
