# Peer review of "Adverse Childhood Experiences Predict the Phenome of Affective Disorders and These Effects Are Mediated by Staging, Neuroimmunotoxic and Growth Factor Profiles"

_cells, 2022, doi:10.3390/cells11091564_

Round 1
Reviewer 1 Report
Dear authors,
it was a pleasure for me to read your manuscript.
The article has very-good quality; in fact, it is original, well thought out, and informative. I think the last figure (fig. 8), which summarizes the results of the study, is very useful. I appreciated your choice to structure the discussion in subsections, making it more readable.
I suggest some concepts that could help strengthen the manuscript quality, which I list below:
-you should add a short section explaining the objectives of the study and a short section showing the study's limitations (e.i. the study was conducted on a small sample of 20 healthy controls and 30 depressed patients).
-introduction, at page 2, lines 55-59: the concept 7 needs to be clarified
-clinical measurements, at page 4, lines 180-184: you should change the font size to make it similar to the one used in the rest of the article
-tables 1 and 2 should be moved from pages 5-6 to page 8, in the results section.
-all abbreviations/acronyms used in tables and figures should be defined in the table note or figure caption, respectively, even though the abbreviations/acronyms were also defined in the text. You should explain the abbreviation/acronym, shown in their tables, in a footnote. In the caption of the table, the abbreviations/acronyms (and their definition) should be separated from each other by a semicolon. For example: in table 1 you should add the definition of TUD, ACE, FEPT, df; in table 2 UNST and STIM; in table 7 DEP; in table 9 ID and FDR
- in the description of the tables you should report the significance value p. To make the table more readable, you could bold the p-values that are significant from your analysis
- results, at page 8, line 343, I can read that table 3 “showed significant group X time interactions for 15 cytokines/growth factors”, but the table shows 16 variables
-the quality of figure 6 is not satisfactory. You should replace the figure with a larger, better resolution figure.
Author Response
It was a pleasure for me to read your manuscript.
The article has very-good quality; in fact, it is original, well thought out, and informative. I think the last figure (fig. 8), which summarizes the results of the study, is very useful. I appreciated your choice to structure the discussion in subsections, making it more readable.
I suggest some concepts that could help strengthen the manuscript quality, which I list below:
-you should add a short section explaining the objectives of the study and a short section showing the study's limitations (e.i. the study was conducted on a small sample of 20 healthy controls and 30 depressed patients).
@ANSWER: added to the Introduction:
Toward this end we employed a precision nomothetic approach [3] including Partial least Squares analysis to delineate the causal effects of ACEs on ROI and immune activation and the cumulative effects of those predictor variables on the affective phenome. The precision nomothetic approach [3] was also used to delineate new pathway phenotypes [3] based on ACEs, ROI, immune activation and the affective phenome.
And we added a limitation section. It reads:
The current study's findings should be discussed in light of its limitations. First, this study would have been more interesting if we also had measured biomarkers of oxidative and nitrosative stress as well as other growth factors and inflammatory mediators. Second, although well powered, the study was conducted on a smaller sample of 20 healthy controls and 30 depressed patients.
-introduction, at page 2, lines 55-59: the concept 7 needs to be clarified
@ANSWER: MDE is now clarified as: ]. Additionally, during episodes of unipolar or bipolar major depressive disorder, named major depressive episodes (MDEs),
-clinical measurements, at page 4, lines 180-184: you should change the font size to make it similar to the one used in the rest of the article
@ANSWER: OK, done
-tables 1 and 2 should be moved from pages 5-6 to page 8, in the results section.
@ANSWER: moved to page 8
-all abbreviations/acronyms used in tables and figures should be defined in the table note or figure caption, respectively, even though the abbreviations/acronyms were also defined in the text. You should explain the abbreviation/acronym, shown in their tables, in a footnote. In the caption of the table, the abbreviations/acronyms (and their definition) should be separated from each other by a semicolon. For example: in table 1 you should add the definition of TUD, ACE, FEPT,
@ANSWER: all added to table 1
df; in table 2 UNST and STIM;
@ANSWER: all added to Table 2
in table 7 DEP
@ANSWER: added to Table 7
in table 9 ID and FDR
@ANSWER: all added to table 8
- in the description of the tables you should report the significance value p. To make the table more readable, you could bold the p-values that are significant from your analysis
@ANSWER: all significant p values are shown in bold (except for the annotation analyses as we only show the most significant terms).
- results, at page 8, line 343, I can read that table 3 “showed significant group X time interactions for 15 cytokines/growth factors”, but the table shows 16 variables
@ANSWER: changed into 16, indeed
-the quality of figure 6 is not satisfactory. You should replace the figure with a larger, better resolution figure.
@ANSWER: we now show another Figure 6 which is better.
Reviewer 2 Report
The manuscript entitled “Adverse childhood experiences predict the phenome of affective disorders and these effects are medicated by staging, neuroimmunotoxic and growth factor profiles” presents the comparison of cytokines and growth factor measurements from the blood of healthy and depressed patients. The results were well analyzed and summarized.
- line 191. Why was the whole blood culture supernatant stimulated? I understand it was done to compare the immune response reactivity. But why was PHA and LPS used and what is the clinical meaning of these stimulants?
- Could the authors give more details of the clinical meanings of the findings. In example, as in line 621, what is the relationship of FDF and depression?
- There are some acronyms not with the full words. Some of them are PHA, PON1, CV, VIF. I can guess, but it may be better to clearly indicate the meanings.
Author Response
The manuscript entitled “Adverse childhood experiences predict the phenome of affective disorders and these effects are medicated by staging, neuroimmunotoxic and growth factor profiles” presents the comparison of cytokines and growth factor measurements from the blood of healthy and depressed patients. The results were well analyzed and summarized.
line 191. Why was the whole blood culture supernatant stimulated? I understand it was done to compare the immune response reactivity. But why was PHA and LPS used and what is the clinical meaning of these stimulants?
@ANSWER: This is now addressed in the methods section as:
Whole blood culture supernatants, both stimulated and unstimulated, were used because this method allows to assay cytokines or growth factors which are otherwise difficult to measure in serum or plasma, including IL-5, IFN-γ, IL-2 and IL-15. Moreover, lipopolysaccharide (LPS) + phytohemagglutinin (PHA) stimulated cultures were used because these measurements reflect the in vivo cytokine production [49-51]. Moreover, the LPS+PHA stimulated production of cytokines and growth factors reflect the capacity to respond to polyclonal activators reflecting the responsivity of the immune system to bacterial and viral infections [49-51].
Could the authors give more details of the clinical meanings of the findings. In example, as in line 621, what is the relationship of FDF and depression?
@ANSWER: This is addressed in the text as:
This is important because the subnetwork of the growth factors measured here interacts with the cytokine network thereby contributing to immune responsivity and immune activation via different pathways as described in section 4.3.
There are some acronyms not with the full words. Some of them are PHA, PON1, CV, VIF. I can guess, but it may be better to clearly indicate the meanings.
@ANSWER: all are now explained in the text as: phytohemagglutinin (PHA); PON1: paraoxonase 1: CV coefficient of variation; VIF à variance inflation factor